# Alterations of aqueous humor Aβ levels in Aβ-infused and transgenic mouse models of Alzheimer disease

Da Eun Kwak[1,2☯], Taeho Ko[1,3☯], Han Seok Koh[4¤], Yong Woo Ji[4,5], Jisu Shin[1,2], Kyeonghwan Kim[1,2], Hye Yun Kim[1,2], Hyung-Keun Lee[4]*, YoungSoo Kim[1,2,3,6]*

1 Department of Pharmacy, Yonsei University, Incheon, Republic of Korea, 2 Yonsei Institute of Pharmaceutical Sciences, Yonsei University, Incheon, Republic of Korea, 3 Industrial Pharmaceutical Sciences, Yonsei University, Incheon, Republic of Korea, 4 Department of Ophthalmology, Institute of Vision Research, Yonsei University College of Medicine, Seoul, Republic of Korea, 5 Department of Ophthalmology, National Health Insurance Service Ilsan Hospital, Goyang, Republic of Korea, 6 Integrated Science and Engineering Division, Yonsei University, Incheon, Republic of Korea

☯ These authors contributed equally to this work.
¤ Current address: Biorchestra, Yuseong-gu, Daejeon, Republic of Korea
* shadik@yuhs.ac (HKL); y.kim@yonsei.ac.kr (YSK)

**Data Availability Statement:** All relevant data are within the paper and its Supporting Information files.

## Abstract

Alzheimer's disease (AD) is an ageing-related neurodegenerative disease characterized and diagnosed by deposition of insoluble amyloid-β (Aβ) plaques in the brain. The plaque accumulation in the brain directly affects reduced levels of Aβ in cerebrospinal fluid (CSF) and blood, as Aβ can freely transport the blood-brain barrier, and clinical investigations have suggested these two biofluids as promising samples for *in vitro* diagnosis. Given that the human eye structurally resembles the brain and Aβ accumulation often observed in the ocular region of AD patients, in this study, we examined aqueous humor Aβ as another possible surrogate biomarker. First, using the acute Aβ-infused AD mouse model by injecting Aβ to the CSF in intracerebroventricular region of normal ICR mice, we investigated whether Aβ concentration in the aqueous humor in AD models is positively correlated with the concentration in the CSF. Then, we examined the correlation of aqueous humor Aβ levels with increased plaque deposition in the brain and reduced Aβ levels in both CSF and blood in adult and aged 5XFAD Alzheimer transgenic mice. Collectively, the synthetic Aβ injected into CSF immediately migrate to the aqueous humor, however, the age-dependently reducing pattern of Aβ levels in CSF and blood was not observed in the aqueous humor.

## Introduction

Abnormally increased production and deposition of the amyloid-β (Aβ) peptide in human nervous system is a typical characteristic of Alzheimer disease (AD) [1]. During the pathological progression of AD, the amyloid precursor protein (APP) on the membrane of neurons is sequentially cleaved by β- and γ-secretases and releases excessive Aβ to the extracellular regions. Although the human brain has efficient clearance systems to remove toxic Aβ such as

**Funding:** This work was supported partially by National Research Foundation (Basic Science Research Program NRF- 2018R1A6A1A03023718 and 2018R1D1A1B0704885 and Original Technology Research Program for Brain Science NRF-2018M3C7A1021858) and partially by the Korean Health Technology R&D Project through the Korea Health Industry Development Institute (KHIDI) funded by Ministry of Health & Welfare, Republic of Korea (HI18C1159).

**Competing interests:** The authors have declared that no competing interests exist.

protein degradation, blood-brain barrier (BBB) efflux, glymphatic system clearance, and meningeal lymphatic vessel transport [2], the Aβ peptide in high concentration shows misfolding behaviors and begins to accumulate in the brain of AD patients, even before the onset of cognitive deficits [3]. Unfolded monomeric Aβ is reported to participate in the physiological synaptic processes [4].

The definitive diagnosis of AD has required the detection of Aβ deposits in the brain either by biopsy, autopsy, or positron emission tomography along with the signs of neurodegeneration [5]. Recently, cerebrospinal fluid (CSF) Aβ(1–42) was suggested as an alternative biomarker for the amyloid concentration measurement by the 2018 revision of AD diagnostic criteria by National Institute on Aging and Alzheimer's Association (NIA-AA) [5–8]. Measurements of CSF Aβ(1–42) show high diagnostic accuracy [9, 10]. It is notable that, while Aβ concentration increase and soluble oligomers and insoluble plaques build up in the brain, the alteration of Aβ levels in CSF shows a proportionally inverse behavior. The concentration of CSF Aβ(1–42) in AD patients is reduced compared to normal adults, inversely indicating the Aβ deposition in the brain [11–13]. Despite the stacked evidence, CSF Aβ(1–42) level is not routinely used in the clinical AD cases due to the complicated sample collection procedure [14, 15]. Clinical investigations searching for the less invasive biomarkers focused on blood Aβ for its clear BBB transporting mechanism through low density lipoprotein receptor-related protein 1 [16] and, thus, anticipated role to directly reflect the Aβ alterations in CSF. Since the analytical results have the discrepancy between the studies [17, 18], the usage of plasma Aβ(1–42) level as a biomarker has not been consolidated in medical practice [19]. It is attributed to the systemic circulation nature of plasma, where the protein level can be affected by the peripheral cleavage of APP or vascular risk factors [20–22]. Therefore, it is essential to explore the novel biofluid to accurately reflect the pathologic changes of AD.

The human eye has neural similarities with the brain containing high-density of neurons and glia cells and has blood barrier [23]. Given the shared functional and structural features of brain and ocular tissues, it is no surprise that the eye has been studied as a window of the brain [24]. Previously, the lens and retina regions were reported to excessively produce Aβ and show accumulation of the soluble and insoluble aggregates of the peptide [25–31]. For the ease of *in vitro* diagnosis, among many ocular regions, we focused on the eye fluid, the aqueous humor, in the anterior chamber [23, 32]. Aqueous humor shares similar characteristics with CSF and plasma to contain a complex mixture of proteins [33]. As the eye lack efficient amyloid clearance systems compared to the brain, the aqueous humor might not directly mirror Aβ level alteration in CSF [27]. A previous study reported that the higher Aβ(1–40) level was detected in aqueous humor of AD patients [26].

In this study, we examined the potent surrogate biomarker role of the aqueous humor Aβ (1–42) to reflect the AD manifestation. To investigate the correlation of Aβ(1–42) level in the aqueous humor with that in the brain, the CSF, and the blood, we conducted a series of *in vivo* experiments using two, Aβ-infused and transgenic (TG), Alzheimer mouse models. The Aβ-infused mouse model bypasses the ageing and APP processing steps and allow us to control the region-specific concentration changes of Aβ. After the injection of monomeric Aβ(1–42) directly into the intracerebroventricular (ICV) of the mouse brain in time- and dose-dependent manner, we measured levels of CSF, blood plasma, and aqueous humor Aβ(1–42) to examine if Aβ is transported from CSF to the aqueous humor. To further investigate the surrogate biomarker role of aqueous humor Aβ, we used the 5XFAD TG mouse model expresses human Aβ in its central and peripheral nervous system. We collected brain, CSF, blood plasma, and aqueous humor samples of adult and aged 5XFAD in both male and female genders and compared changes of Aβ(1–42).

## Materials and methods

### Animals models

Transgenic mouse (strain name; B6SJL-Tg(APPSwFlLon,PSEN1*M146L*L286V) 6799Vas/ Mmjax) carrying five mutations associated with early onset familial Alzheimer's disease (FAD) was used in the experiment. The 5XFAD mice were obtained from Jackson Laboratory (USA) and have been maintained by mating with C57BL/6 X SJL wild type mice. Institute of Cancer Research (ICR) mice (strain name; Crl:CD1, male, six-week-old) were purchased from Orient-bio Inc. (Seoul, Korea). The strain is a fertile albino mouse that is widely used for the disease modeling studies. All mice were bred in a laboratory animal breeding room at Yonsei University (Seoul, Korea). They were housed in groups of five per cage with a controlled temperature, humidity, and a 12/12 hour light/dark cycle. Water and food were available *ad libitum*. All animal experiments were carried out in accordance with the National Institutes of Health (NIH) Guide for the Care and Use of Laboratory Animals. The research protocol was approved by the Institutional Animal Care and Use Committee of Yonsei University, Seoul, Korea (IACUC-A-201806-744-01).

### ICV injection of Aβ(1–42) peptide

Aβ(1–42) peptides were synthesized using solid-phase peptide syntheses as previously reported [34]. Synthetic Aβ(1–42) peptides were dissolved in 10% dimethyl sulfoxide (DMSO) in distilled water at 0.5, 1, 2, and 4 nmol (5 μL of 100, 200, 400, and 800 μM). The mice were anesthetized with 4% avertin by intraperitoneal injection. Aβ(1–42) solutions were injected into the cerebral ventricle of mouse brain according to the previously reported protocol [35]. The injection site was 1.0 mm posterior to bregma, 1.8 mm lateral to the sagittal suture, and 2.4 mm in depth. Hamilton syringe with a 26-gauge stainless-steel needle was used to inject the Aβ(1–42) solutions.

### Intravenous (IV) injection of Aβ(1–42) peptide

The 26.5-gauge syringe was prepared for Aβ(1–42) injections. To clearly see the lateral veins on both sides of a tail, the heat was applied to make the veins dilated using a 200 W lamp. Since the mice were not anesthetized, the restraining device was required to gain access to the mice veins. The mice were given stress until their lateral veins visible, followed by the administration of 100 μL of 40 μM Aβ(1–42) diluted in 1X phosphate-buffered saline (PBS).

### Collection and sample preparation of brain, CSF, aqueous humor, and plasma

To obtain CSF and aqueous humor samples from mice, PYREX glass capillary tubes with a diameter of 1.5 mm were used. The capillary tube was flame-polished to obtain a diameter of 0.5 mm. The capillary tube was used to collect CSF from cisterna magna [36]. Then, we obtained aqueous humor with inserting capillary tubes to the center of cornea, enabling to reach the anterior chamber [37]. All mice were sacrificed by cervical dislocation after sampling. Collected CSF and aqueous humor samples were frozen immediately. A blood sample from the vena cava was transferred to EDTA tube and was centrifuged (3,000 rpm, 15 minutes, 4˚C) to separate plasma. Protease inhibitor cocktail (Roche Diagnostics, Switzerland, cat# 11836170001) was then added to the plasma. The CSF, aqueous humor, and plasma samples were stored at −80˚C in the freezer. For cryosection of the brain, each brain was initially fixed in 4% paraformaldehyde (pH 7.4) and was transferred to 30% sucrose after 24 hours. Then, the brain was cut into 35-μm-thick slices using a Cryostat (Leica, CM1860).

## Analysis of Aβ(1–42) levels by sandwich-ELISA

Levels of Aβ(1–42) in biofluids were quantified by using human Aβ(1–42) ultrasensitive ELISA kit (Invitrogen, cat# KHB3544). CSF and plasma samples were 1,000-fold diluted. In time-dependent measurements, aqueous humor samples were 200-fold diluted. To detect the differences in low injection concentrations, aqueous humor samples were diluted 100-fold in dose-dependent measurements. When analyzing CSF, aqueous humor and plasma samples of 5XFAD were diluted 100-, 15-, and 5-fold respectively. The sandwich-ELISA was performed according to the manufacturer's instructions using the diluted samples.

## Immunohistochemistry assay

Brain slides were washed with 1X PBS 3 times, 5 minutes each, followed by the antigen was retrieved using 1% SDS in PBS for 10 minutes. The slides were washed with PBS, and 20% horse serum in PBS was used as a blocking reagent. We incubated the slides with 6E10 antibody (1:200, Covance) at 4˚C, overnight. Then, the slides were incubated with goat anti-mouse IgG conjugated with Alexa Fluor Plus 488 (1:200, Life Technologies) for 1 hour at room temperature. Image were taken using a Leica DM2500 fluorescence microscope. The number of amyloid plaques was quantified using ImageJ software.

## Statistical analysis

Statistical analysis was conducted with GraphPad Prism 7 using Student's unpaired t-test comparisons and repeated-measures analysis of one-way ANOVA, followed by Tukey's post hoc comparisons ($^*P < 0.05$, $^{**}P < 0.01$, $^{***}P < 0.001$, $^{****}P < 0.0001$; other comparisons were not significant). Data were presented as mean ± SEM of each group.

## Results

### Time-dependent transport of Aβ(1–42) from CSF to aqueous humor

APP are on the membrane of neurons and, thus, their enzymatic cleavages releasing Aβ peptides are found in both central and peripheral nervous systems [38]. To eliminate the possibility that the Aβ found in the aqueous humor is produced in the eye, instead of being transported from the brain, we used Aβ-infused AD mouse. Previously, we reported an *in vivo* technique to acutely induce Alzheimer-like symptoms by ICV injection of Aβ [35]. This model is a useful tool to investigate Aβ-dependent pathology of AD by allowing researchers to control amyloid in a region-, a time-, and a dose-dependent manner. To verify that Aβ(1–42) migrates from the brain to the aqueous humor, 4 nmol of Aβ(1–42) was injected into the ICV regions of the brain of 6-week-old normal male ICR mice. We prepared five groups of Aβ-infused mice (male, n = 3 per group) each for separate time points and, in 15, 30, 60, 120, and 240 minutes since the ICV injection, we collected aqueous humor samples. We then used human Aβ(1–42) ultrasensitive ELISA kits to measure the biomarker concentration in each sample with triplicates (**Fig 1A**). As a result, in the aqueous humor samples of the subject mice, the artificially injected synthetic Aβ(1–42) was detected with the maximal peptide concentration at 30 minutes (**Fig 1B**). This finding is consistent with previous studies reporting the half-time of Aβ efflux from CSF to blood to be 34.63 minutes [34, 39].

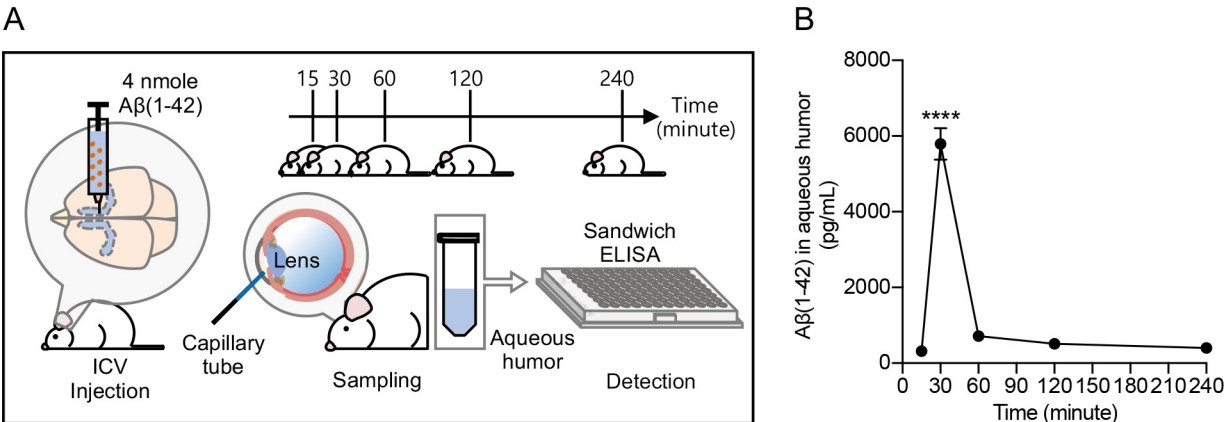

**Fig 1. Time-dependent measurements of Aβ(1–42) levels in aqueous humor after ICV injection.** ICR mice were prepared to make acute AD model (male, n = 3 per group). (**A**) Upper: the scheme of time-dependent experiment of aqueous humor sampling (15, 30, 60, 120, and 240 minutes). Left: ICV injection of Aβ(1–42) 4 nmol. Center part: aqueous humor collection using a capillary tube. Right: the measurement of Aβ(1–42) concentration by sandwich-ELISA. (**B**) Concentrations of Aβ(1–42) in aqueous humor were analyzed by ELISA. The data was analyzed by one-way ANOVA followed by Tukey's post hoc comparisons tests. (****P < 0.0001).

## Dose-dependent measurements of Aβ(1–42) levels in the aqueous humor upon ICV Aβ injection

Interim results support our hypothesis that Aβ(1–42) can transport from CSF to the aqueous humor in the mouse model. However, it is still uncertain whether the Aβ(1–42) level in the aqueous humor reflects that in CSF and how this biomarker transports from CSF to the aqueous humor. Although the blood is suspected, the migration route from CSF to aqueous humor is unclear yet. To assess the correlation of Aβ(1–42) levels in the aqueous humor with those in CSF and blood, the Aβ(1–42) peptide in various concentrations (0.5, 1, 2, and 4 nmol) was injected into the ICV region of normal ICR mice (male, n = 5 per each dose group). At the peak time of the maximal Aβ(1–42) in aqueous humor, 30 minutes from ICV injection, we collected CSF, blood, and aqueous humor samples of each (**Fig 2A**). First, the increase of CSF Aβ(1–42) levels was confirmed in a dose-dependent manner and the result supports that the ICV injection of Aβ(1–42) was successfully performed (**Fig 2B**). Secondly, as the concentration of injected Aβ(1–42) in CSF increased, the Aβ(1–42) levels in aqueous humor also increased (**Fig 2C**). This result indicates that acute changes in the concentration of Aβ(1–42) in the CSF can be reflected in the aqueous humor. In addition, we observed that the Aβ(1–42) level in the blood plasma also elevated depending on the increase concentration of injected peptide in the ICV (**Fig 2E**).

To investigate the migration route of Aβ, we injected the synthetic Aβ(1–42) intravenously into the tail vein and collected aqueous humor samples with various time intervals (5, 15, 30, and 60 minutes) since the injection. As a result, we observed that the Aβ(1–42) levels in plasma significantly decreased over the time, which is consistent to the previous plasma and serum stability study [40] (**Fig 2F**). Aβ(1–42) levels in aqueous humor also decreased in the time-dependent manner which is similar to the plasma result. However, the progression was slower and Aβ(1–42) were highly detected in aqueous humor of 5 and 15 minutes group compared to the blood case of 30 and 60 minutes group (**Fig 2G**). Thus, the migration route of Aβ from the CSF to the aqueous humor is probably through the blood stream and related Aβ influx/efflux systems such as receptor for advanced glycation end products and low-density lipoprotein receptor-related protein [41].

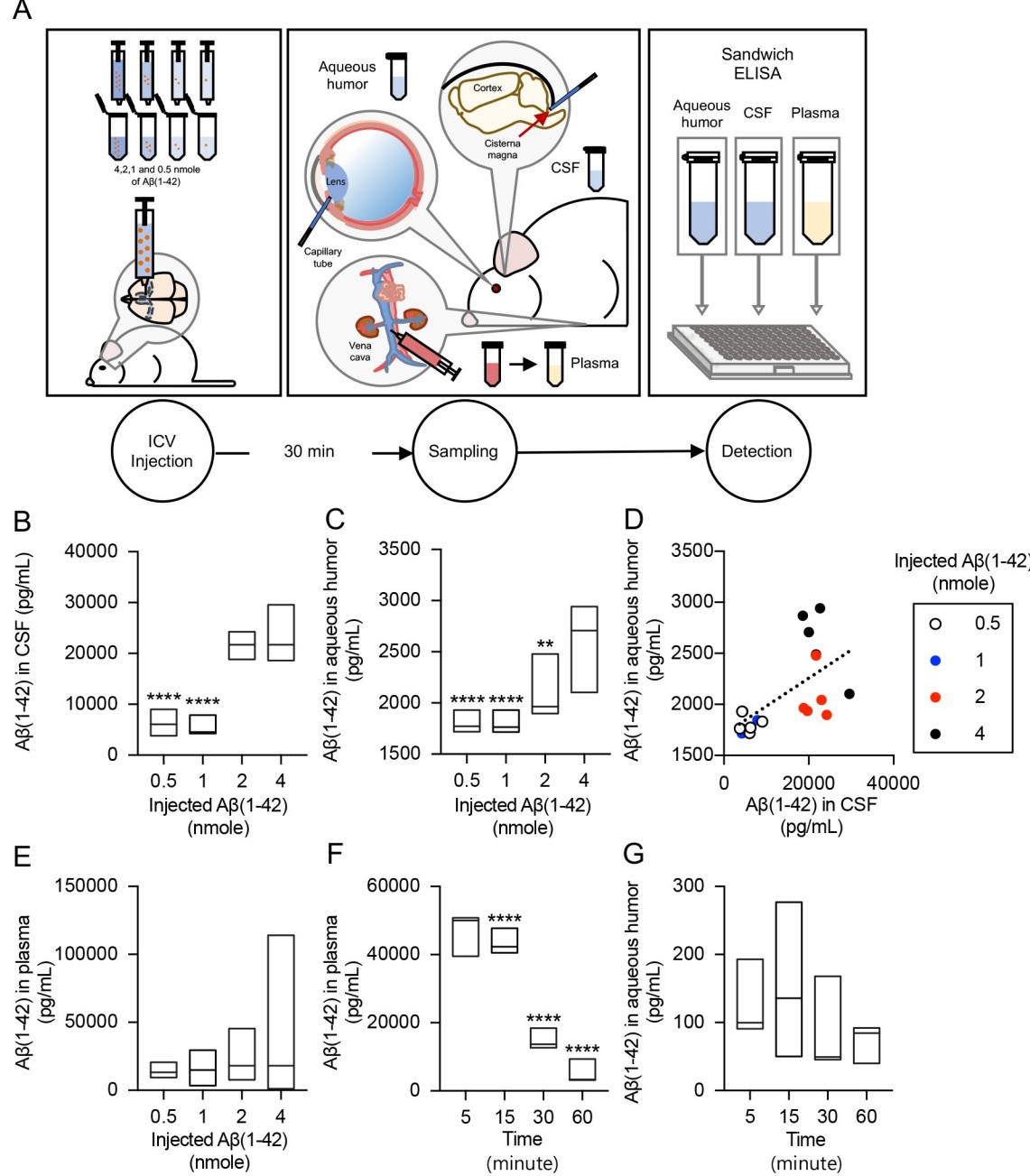

**Fig 2. Dose-dependent measurements of Aβ(1–42) levels in aqueous humor, CSF and blood plasma after ICV injection.** ICR mice were prepared to make acute AD model (male, n = 5 per group). **(A)** Left: ICV injection with various Aβ(1–42) concentrations (0.5, 1, 2, and 4 nmol). Center: the collection of CSF, aqueous humor, and plasma at 30 minutes after ICV injection. Right: the analysis of biofluids Aβ(1–42) using sandwich ELISA. Aβ(1–42) levels in each **(B)** CSF, **(C)** aqueous humor, and **(E)** plasma were shown. **(D)** Scatter plot data of Aβ(1–42) levels in CSF and aqueous humor. The color of each circle represents the injection concentration of Aβ (1–42). Aβ(1–42) was analyzed in **(F)** plasma and **(G)** aqueous humor samples obtained at various time intervals (5, 15, 30, and 60 minutes) after IV injection. Data is presented as mean ± SEM against the highest concentration group in all biofluids. Standard deviation values for each group are listed in the supporting information (**S1 Table**). Significance was tested by one-way ANOVA followed by Tukey's post hoc comparisons tests. (**P < 0.01, ***P < 0.001, ****P < 0.0001).

## Ageing- and gender-dependent alterations of Aβ(1–42) in brain, CSF, blood, and aqueous humor

The Aβ-infused mice utilized in the former set of experiments can be useful when the study needs to control Aβ and bypass its upstream cascades. However, in most pathophysiological phenomenon, the Aβ-infused model is less close to human AD cases compared to transgenic models. Thus, transgenic mice with human APP mutation genes are useful models to investigate amyloid cascade and related pathology of AD patients. First, to test our hypothesis that Aβ can be detected in the aqueous humor, we extracted eyes of 5XFAD mice and measured the concentrations of Aβ(1–42) in aqueous humor samples in an age-dependent manner. In this experiment, following sample sizes were used for the various age groups: 3.7-month-old (n = 5) for adult female mice, 14-month-old (n = 8) for aged female mice, 5-month-old (n = 5) for adult male mice, and 12-month-old (n = 10) for aged male mice. Each group was classified according to the disease progression. The tendency was analyzed by human Aβ(1–42) ultra-sensitive ELISA kits between the levels of Aβ(1–42) detected in biofluids by gender and age differences. To determine whether Aβ peptide is progressively accumulated in the brain during the aging, immunohistochemistry assay was performed using an antibody capable of specifically detecting Aβ (**Fig 3A**). Compared to the adult mice, the higher levels of Aβ aggregates were found in the brains of aged mice (**Fig 3B**). In the CSF, Aβ(1–42) concentration decreased as ages increased (**Fig 3C**). Plasma Aβ(1–42) level also showed a decreasing tendency along the ages (**Fig 3D**). In consistent with previous reports in AD-related transgenic mouse models, the decrease of Aβ(1–42) levels in CSF and plasma was reproduced in this study [42]. Interestingly, the Aβ(1–42) levels in aqueous humor was not decreased with ages. In the female, particularly,

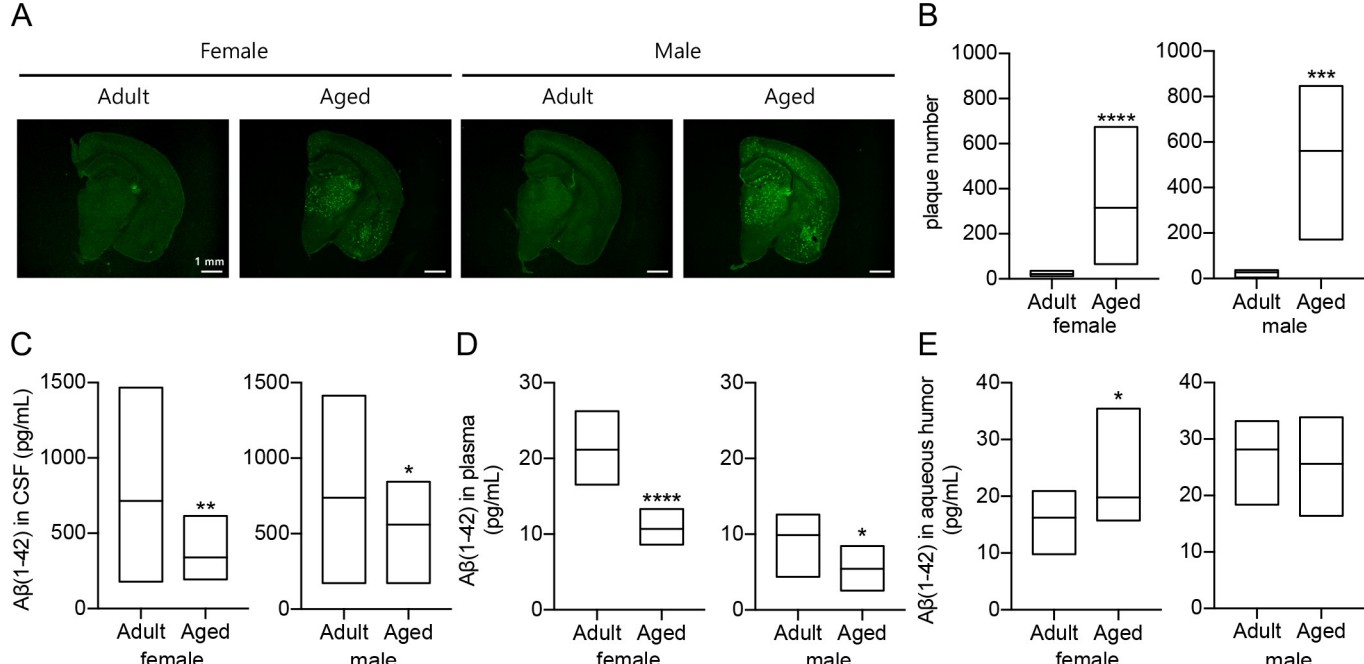

**Fig 3. Comparison of Aβ(1–42) levels in brain, CSF, plasma, and aqueous humor between adult mice and aged mice.** Analysis of Aβ levels in the brain and biofluids (CSF, plasma, and aqueous humor) in male and female. (n = 5 for 3.7-month-old TG female; n = 5 for 5-month-old TG male; n = 8 for 14-month-old TG female; n = 10 for 12-month-old TG male). **(A)** Representative brain hemisphere images stained with 6E10 antibody with age and gender differences. Scale bar = 1 mm. **(B)** Comparisons of the plaque numbers between adult and aged TG mice in each gender. The analysis of Aβ(1–42) levels in **(C)** CSF, **(D)** plasma, and **(E)** aqueous humor samples was individually shown. The differences were analyzed by unpaired t-test. (*P < 0.05, **P < 0.001, ***P < 0.001, ****P < 0.0001).

the levels of Aβ(1–42) in aged 5XFAD mice was increased compared with those in adult 5XFAD mice. Moreover, the two male groups did not show any significant difference in Aβ(1–42) levels (**Fig 3E**).

## Discussion

Collectively, in AD animal mice, we observed that (1) Aβ(1–42) monomers infused into CSF in the ICV region acutely transport to the blood and the aqueous humor, (2) artificially injected Aβ(1–42) levels in CSF are proportionally reflected in the aqueous humor in mice without human APP expression, and (3) the progressive decrease of Aβ(1–42) levels in representative fluid biomarkers, CSF and blood, is not observed in aqueous humor. In this study, we infused the Aβ(1–42) monomer into CSF by ICV injection in normal ICR mice, thereby observed the corresponding increase in the aqueous humor. This acute AD model showed that ICV injected Aβ reaches the aqueous humor by excluding the other possible origins of Aβ detected in aqueous humor. As a result, the level of Aβ(1–42) in aqueous humor reached a peak concentration after 30 minutes of ICV injection. Moreover, as the injection concentration increased, detected Aβ(1–42) level in aqueous humor has also displayed an increasing trend.

We inferred the possible migration route of Aβ was through the blood. The Aβ peptide generated in the brain can be released into blood by the low-density lipoprotein receptor-related protein [43]. Aqueous humor is secreted from the ciliary body, the circulation of which produces aqueous humor by blood ultrafiltration [44]. As injecting Aβ(1–42) to the normal ICR mice intravenously, spiked Aβ concentration was reflected in aqueous humor. We concluded that the Aβ monomer passes blood-brain barrier and ultimately enters the aqueous humor. The exact molecular mechanism of how Aβ that was originally in the CSF can be detected in the aqueous humor needs further investigation.

Through examining the aqueous humor of adult and aged 5XFAD TG mice with similar AD pathophysiology to human, we found the Aβ(1–42) level elevation in the aqueous humor can be a surrogate biomarker for the AD progression. The tendency of the aqueous humor was different with that of CSF or plasma, both of which showed decreasing trend along the ages. It can be presumed that the amount of Aβ(1–42) in the aqueous humor is more derived from APP peripheral cleavage in the ocular tissues than that from the CSF or plasma [33, 45]. In female TG mice, Aβ(1–42) levels of the aqueous humor were more closely associated with the amyloid plaque burden in the brain rather than those of the CSF, independent of the migration that occurs. Unlike the clear tendency in females, no significant differences between adult and aged mice were identified in males. This result may be due to the gender-based difference of composition ratio in the aqueous humor proteome [46].

Regarded as ocular AD, primary open-angle glaucoma (POAG) shares similar underlying etiology with AD, both of which are age-related and cause neurodegeneration [47–50]. The elevation of intraocular pressure (IOP) caused by the imbalanced flow of aqueous humor is a key risk factor of POAG. The neuronal cell deaths often continue to occur even after modulating IOP to normal levels, and Aβ is likely to mediate the development of retinal ganglion cells (RGC) apoptosis which implicates neurotoxic effect [38, 51]. Since these ophthalmic biochemical changes occur earlier than the onset of AD, aqueous humor Aβ(1–42) can be analyzed in the preclinical stage of AD [52–54].

As a preliminary stage, we revealed the link of Aβ(1–42) level between CSF and aqueous humor in acute AD mouse. Intriguingly, the propensity of aqueous humor Aβ(1–42) level in TG mice was a bit different from the CSF Aβ(1–42) level. The limitation of our research is a lack of comparison with the non-AD having genetically expressed APP protein, due to the

experimental disease model displaying only human Aβ(1–42). We suggest the role for aqueous humor Aβ(1–42) that indirectly reflecting the AD-related pathology. Our future direction would be the longitudinal study in clinics using aqueous humor samples to measure Aβ(1–42) levels and observe the incidence of AD. A critical factor for this approach is how to collect the aqueous humor from the patients. Seeing the increasing frequency of relatively common ophthalmic surgeries, aqueous humor samples could be easily obtained through the surgeries in elderly population [55, 56].

## Supporting information

**S1 Table. Statistical analyses of data in Figs 1, 2, and 3.**
(TIF)

## Acknowledgments

All images are created by authors for this paper.

## Author Contributions

**Funding acquisition:** Yong Woo Ji, Hyung-Keun Lee, YoungSoo Kim.

**Investigation:** Da Eun Kwak, Taeho Ko, Han Seok Koh, Jisu Shin, Kyeonghwan Kim.

**Project administration:** Yong Woo Ji, Hyung-Keun Lee, YoungSoo Kim.

**Resources:** Jisu Shin.

**Supervision:** Hyung-Keun Lee, YoungSoo Kim.

**Writing – original draft:** Da Eun Kwak, Taeho Ko, Hyung-Keun Lee, YoungSoo Kim.

**Writing – review & editing:** Hye Yun Kim.

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
