## [Decision Letter · Decision Letter 0]

4 Oct 2019

PONE-D-19-24249

Correlations of amyloid-β levels between CSF and aqueous humor in transgenic and acute Alzheimer disease mouse models

PLOS ONE

Dear Professor Kim,

Thank you for submitting your manuscript to PLOS ONE. After careful consideration, we feel that it has merit but does not fully meet PLOS ONE’s publication criteria as it currently stands. Therefore, we invite you to submit a revised version of the manuscript that addresses the points raised during the review process.

All the critical issues have been raised by the Reviewers, please address all the points.

We would appreciate receiving your revised manuscript by Nov 18 2019 11:59PM. To enhance the reproducibility of your results, we recommend that if applicable you deposit your laboratory protocols in protocols.io, where a protocol can be assigned its own identifier (DOI) such that it can be cited independently in the future. For instructions see: http://journals.plos.org/plosone/s/submission-guidelines#loc-laboratory-protocols

We look forward to receiving your revised manuscript.

Kind regards,

Madepalli K. Lakshmana, Ph.D

Academic Editor

PLOS ONE

Journal Requirements:

Reviewers' comments:

Reviewer's Responses to Questions

**Comments to the Author**

1. Is the manuscript technically sound, and do the data support the conclusions?

Reviewer #1: No

Reviewer #2: Yes

Reviewer #3: Yes

2. Has the statistical analysis been performed appropriately and rigorously? 

Reviewer #1: No

Reviewer #2: No

Reviewer #3: No

3. Have the authors made all data underlying the findings in their manuscript fully available?

Reviewer #1: No

Reviewer #2: Yes

Reviewer #3: No

4. Is the manuscript presented in an intelligible fashion and written in standard English?

Reviewer #1: Yes

Reviewer #2: Yes

Reviewer #3: Yes

5. Review Comments to the Author

Reviewer #1: Synopsis

In this study the authors investigate the relationship of Aβ42 in CSF and aqueous humor fluid (AHF) in two animal models, one double transgenic (dTG) and one acute model by ICV injection of synthetic Aβ42. They show increased levels of Ab42 in AHF of dTG mice at 12 and 24 months of age, they also report a slight increase between the age group. Using ICV model the author show that injected Aβ reaches the AHF in a time- and dose-dependent manner.

Comments:

This could have been a very interesting paper if the authors have had executed the experiment in whole and presented all their data. Currently it appears they just show a selection of the data that support their claim. The data on CSF Aβ and the Aβ deposits in the brain of TG animals is missing. They hence do not show if Aβ is accumulated in AHF in relation to CSF and the brain Aβ (levels/deposits), allowing them to examine if AHF Aβ can be useful as a surrogate marker for AD diagnosis. The ICV experiment is also not complete as they should have tested iv injection of Aβ in circulation to perhaps be able to substantiate the route of Aβ to AHF. These shortcomings allow the authors to make just one single conclusion that ICV injected Aβ can reach the AHF. This is insufficient however to establish that AHF Aβ can be used or is useful as a surrogate biomarker for CSF Aβ for AD diagnosis.

Thus many of conclusions and statements of the authors are either misgiving and or unjustified, as follows below:

1)- in 230-232, as this sentence gives a generalized expression rather than what the authors can claim in this particular experiment. Rephrase perhaps as follows: “This specific animal model showed that ICV injected Aβ reaches the AHF fluid”. You cannot rule out other sources at all as the ICR animals are not TG and your Aβ ELISA is human specific. Even if the animals were TG the experiment could not rule out peripheral sources by mere ICV injection and concentration changes, since even in TG there should be a pre-established equilibrium and ICV Aβ injection just add to this level.

2)- Also the conclusion in line 240-241 has to be carefully revised as it is, it is greatly misgiving. If you inject Aβ in circulation and then measure it in AHF vs CSF, you could perhaps make such a conclusion but as it is this claim is just one of several possibilities. In addition, such an information is useless (in the context of your manuscript) unless you can show that it is accumulated in AHF. Otherwise, why measure Aβ in AHF or CSF (for that matter) if you can measure it in plasma?

3)- The sentence in line 242-244 (“It is assured … TG mice”) is also not correct/justified in the present study. CSF Aβ is a biomarker of AD because of its altered levels in AD vs control and its inverse association with the amount of Aβ deposits in the brain but not merely because Aβ can be measured in CSF (as control subjects have much higher levels of Aβ in the CSF than AD).

4)- Also consider the following shortcoming of the study: You might have been able to provide some clues if you had measured changes in AHF Aβ in relation to CSF Aβ and the Aβ deposits in the brain of the TG animal to establish that levels of Aβ in AHF like that in CSF reflect the extend of Aβ deposits in the brain. Thus currently, the Fig 1 results does not really say much as the comparison between TG vs WT has limited value since your Aβ ELISA can only measure human Aβ. The main interesting point in Fig. 1 is the Aβ level at 12 vs 24 months but since you do not show the levels of plaques and the levels of Aβ in CSF you cannot establish a proper relationship to point out AHF Aβ as a surrogate biomarker.

5)- In Abstract, line 26-28. The author claim that they “… investigated whether the Aβ in the aqueous humor in AD models is directly correlated with the misfolding protein of the central nervous system.” You did not do such an investigation in this study as far as I can deduct from your experiment. As a matter of fact your ICV model reject your statement as you cannot claim (and it is very unlikely) that the synthetic Aβ monomers were all misfolded (not to mention that even in TG or human, one cannot claim that the soluble Aβ peptides are misfolded even though many believe so mainly because of confusion that aggregation can only occur due to misfolding of Aβ peptides and neglect that aggregation may also occur simply because of an excessive accumulation in e.g. parenchymal fluid in the brain.

Other issues:

1. ICR (mice) is not defined in abstract or the rest of the text.

2. Is the Age-dependent increase in AHF Aβ significant? Provide p-values!

3. Why didn’t you measure Aβ in serum, CSF and brain extracts of dTG?

4. To ascertain route of Aβ to AHF, why not also inject Aβ iv in circulation, which is much easier to do and would provide additional information?

5. In Fig 3, plasma Aβ42 at 4 nmole ICV injection is missing.

6. In Fig 3, it is difficult to appreciate the importance/significant of the comparisons with the highest conc group (i.e. 4 nmol group), particularly since there is no difference between 0.5 vs 1 nmol-groups or 2 vs 4 nmol groups (in 3B)? It is important here that direct correlation graphs plus their r- and p-values are shown, as well.

Reviewer #2: Reviewer Comments to Editor:

Recommendation: Accept after major revision.

Reviewer Comments to Author:

Globally, Alzheimer’s disease (AD) is a growing health and economic challenge that has no effective cure. AD is a progressive neurodegenerative disorder and pathological hallmarks include presence of hyperphosphorylated tau and amyloid protein deposition. Currently, these pathological biomarkers are detected either through cerebrospinal fluid analysis or brain imaging. Though effective, these methods are not widely available due to issues such as the difficulty in acquiring samples, lack of infrastructure or high cost associated.

The authors have previously reported a direct correlation of Aβ (1–42) levels between CSF and plasma in AD mouse model by injecting monomeric Aβ (1–42) directly into the intracerebroventricular (ICV) region of normal adult mouse brains to induce AD-like phenotypes. In this manuscript, Kwak et al. reported that presence of Aβ (1-42) in the aqueous humor can be used as a surrogate bio-marker of perturbed Aβ levels in central nervous systems of AD.

This manuscript is well written, concise, easy-to-follow and provides a new insight on the potential bio-markers for AD patients.

I support its acceptance provided the major revisions as noted below are adequately addressed:

Abstract:

• Well-organized and well written.

• Since the word limit is 300, I would suggest authors to include few sentences of background information.

• Preferable to be consistent with either Aβ (1-42) or amyloid beta.

Introduction:

• In the first paragraph, few sentences are written without any citations. Citations would strengthen use point of emphasis.

• It is advisable to revise last paragraph where the author’s discuses about the experiment protocol instead of focusing on the importance of their study.

• Please include one sentence showing the novelty and importance of your study.

Materials and Methods:

• In experiment 2, please elaborate how the aqueous humor at different time point from the same mice were collected.

Results:

• The authors have used male transgenic mice in their experiments whereas several studies (refer to the journals: PMID: 30820070, PMID: 26987699, PMID: 20442496) have reported that the incidence of AD is higher in females. So, I would suggest to repeat the experiments in female mice model to make the data more authenticate to the clinical data.

• Since hyperphosphorylated tau has been found in both AD and Glaucoma, suggesting a possible pathophysiologic link between the two diseases (PMID: 26425322), but no attempt was made to quantify hyperphosphorylated tau in this study.

• The authors measured the Aβ (1–42) concentration in CSF by injecting different concentration of Aβ (1–42) into the ICV region of normal adult ICR mice by following their previous study. But, the result showed significant differences (around 5-folds higher in this study compare to the previous result) between two studies. Why this discrepancy.

• In the description of figure 1 results, the authors claimed that human Aβ (1-42) was not found in the aqueous humor of WT mice, but in figure 1 bar data, it clearly showed the presence of some amount Aβ (1-42) in WT.

• The authors claimed that Aβ transport from the central nervous system to the eye is through blood. So, I would recommend to perform the experiment for Aβ (1–42) concentration in plasma also to see how much Aβ (1–42) transported to the aqueous humor through the blood.

• In figure 2B, the authors measured the half-time of Aβ efflux from CSF to blood and their finding is consistent with the work done by Shibata M. et al. (2000). In addition, Shibata M. et al. (2000) also mentioned that there was also a slow, time-dependent retention of Aβ in brain parenchyma with a t1/2 of 164.5 minutes. But, figure 2B shows after 34.63 minutes, release of Aβ is suddenly stopped from the brain which is not consistent with the Shibata study, i.e. there is no time-dependent retention of Aβ in brain.

• For figure 2B & 3C, the authors followed the same procedure (injecting monomeric Aβ (1–42) directly into the intracerebroventricular (ICV) region of normal adult mouse brains), used same concentration (4nmole), same time point (30 minutes). But, why two different Aβ (1-42) concentrations.

• More labels are necessary in Figs 2A & 3A.

Discussion:

• The authors should discuss more about how the Aβ (1–42) is transported into the aqueous humor through blood-brain barrier.

Reviewer #3: This is a generally well written and interesting paper about the significantly increased levels of Aβ in the aqueous humour of transgenic AD mouse model in comparison to WT, and the ability to measure acutely perturbated levels of CSF Aβ in both the plasma and the aqueous humour of ICR mice, demonstrating that Aβ introduced to CSF is transported into aqueous humour.

There are some grammatical errors, as well as some questions regarding the statistics and presentation of your data that I have listed in the sections below.

In general, where you say central nervous system, please change to CSF for better clarity, as you are referring to soluble Aβ(1-42).

Abstract

• Line 25: “Based on the clinical evidence…”

• Line 27: Please clarify what specifically is correlated; i.e. presence/concentration of Aβ in aqueous humour is positively correlated with the presence/concentration if Aβ in CSF

• Line 31: Consider changing wording, i.e. “To investigate the correlation between Aβ levels in CSF and aqueous humour, we…”

• Line 37: Consider changing conclusion wording, i.e. “Our results indicate that Aβ peptides are elevated in the aqueous humour in AD mice. In addition to this, acute elevation of Aβ peptides through injection into CSF can be detected in the aqueous humour, suggesting that

Introduction

• Line 47: Other limitations to the amyloid-PET scan e.g. cost, exposure to radiation?

• Line 59: “The eye has been suggested…”

• Section beginning with Line 59 is generally unclear; please clarify why you have included information regarding POAG, i.e. do you mean that Aβ in aqueous has been found to be elevated in POAG, and may play a role in neurodegenerative diseases in the eye, and so measuring Aβ in aqueous humour may be useful for AD as well as other neurodegenerative diseases?

• Line 61: Reference 6 does not directly measure levels of Aβ in aqueous humour of patients, so does not support your statement

• Line 62: Please clarify logic for this sentence, your meaning is unclear. Reference 10 is a review paper that discusses the role of Aβ in AMD, and how changes in Aβ may be involved in the early stages of AMD pathogenesis. Do you mean that changes in Aβ levels in aqeuosu humour may also be seen in the early stages of AD pathogenesis?

• Section beginning with Line 59

• Line 72: “Given that human aqueous humour samples have been shown to contain Aβ…” this statement needs a reference

Methods

• Consider adding section summarising statistical analyses used, for better clarity

• Line 99: “…as previously reported”

Results

• Line 140: Should be “Results”

• Line 150: You have stated that Aβ(1-42) was not found in the aqueous humour of WT mice, and you refer to Figure 1. However, Figure 1 shows that > 500 pg/mL of Aβ(1-42) was found in 12-month-old WT, and approx. 250 pg/mL of Aβ(1-42) was found in 24-month-old WT. In addition to this, you noted on line 125 that the analytical sensitivity of your sandwich-ELISA is < 1.0 pg/mL. Please clarify this discrepancy. Do you mean that Aβ(1-42) was present in significantly lower levels in WT compared to TG model?

• Figure 1: Please state whether you have plotted means ± SD or means ± SEM in the figure caption, include the sample size as a legend and add brackets to indicate which two groups you are referring to for each p-value

• For the data shown in Figure 1, you have used an F-test to determine the difference in Aβ(1-42) levels between WT and TG, which assumes both datasets are normally distributed, and have equal variances. Please include the statistical test for normality, and provide a table summarising the numerical mean ± SD for each group

• Please include the F-value, and degrees of freedom, for each p-value you have given for Figure 1

• Line 153: Please perform a statistical analysis to determine whether levels of Aβ(1-42) are significantly different between TG 12 months and TG 24 months

o If all datasets are normally distributed with equal variance, consider doing a one-way ANOVA for all 4 datasets, and then perform a Tukey post hoc test to determine which means are significantly different from each other amongst all 4 groups

• Line 154: Please change “subjects” to “in ageing TG mice”; also, please reword this sentence, the logic is not clear.

• Line 157: Please change “the eye” to “aqueous humour”.

• Line 187: In the caption for Figure 2, please change “central nervous system” to “CSF” and “eye” to “aqueous humour”

• Line 195: please change “central nervous system” to “CSF” and “eye” to “aqueous humour”

• Line 209: please change to “…that acute changes in the concentration of Aβ(1-42) in the CSF…”

• Line 212: This sentence needs a citation

• Please perform a correlation between levels of Aβ(1-42) in CSF and Aβ(1-42) in aqueous humour to show how strongly these are related, include the resulting r2 value and statistics to support statements regarding the “significant correlation” e.g. line 222, and generate a figure showing this data

• Please include the sample size for each time point as a legend, and state in the figure caption whether you have plotted mean ± SD or mean ± SEM

• In Figure 2, the level of Aβ(1-42) in aqueous humour after 4 nmole ICV injection was approx. 6000 pg/mL, however in Figure 3, the level of Aβ(1-42) in aqueous humour after 4nmole ICV injection appears to be half of this, at approx. 3000 pg/mL. Can you comment on why you think this may be the case, as the protocol was the same?

• For Figure 3, you state you have plotted mean ± SEM, however it appears you have used box plots. Please clarify what you have plotted. If they are box plots, this is a good indication of the spread of your data, but please also provide mean ± SD values in a table.

• You have used a one-way ANOVA to determine whether the levels of Aβ(1-42) in CSF, aqueous humour and plasma are significantly different following different injection concentrations, which assumes that datasets are normally distributed and have equal variances. Please include the statistical tests for normality.

• For the data provided in Figure 3, please report the F-values and degrees of freedom from your one-way ANOVA. Also, please clarify which pairwise comparison was significantly different, either on the figure with brackets, or in a table.

• Please include the data for level of Aβ(1-42) in plasma when 4 nmole of Aβ(1-42) was injected, or provide details regarding why this data point was excluded from Figure 3D

Discussion

• Line 228 should read “Aβ levels in CSF and aqueous humour”

• Line 230: Is this statement accurate? To my understanding, although ICR mouse does not specifically express human APP, this mouse model still expresses endogenous murine APP which may produce Aβ which may be detected by ELISA. In addition to this, your WT control had detectable levels of Aβ in aqueous. Better evidence of the movement of Aβ from CSF to aqueous humour would be the change in Aβ concentration over time, as well as the significant changes in Aβ levels in aqueous when you alter injected CSF Aβ concentrations.

• Line 239: You may also wish to add, to further support your argument, that it is known that aqueous humour is composed of blood plasma (REF), and it is believed that aqueous humour proteins are derived from blood plasma, rather than synthesised locally inside the eye (REF).

• Line 240: “Thus, it is possible…”

• Line 242: Consider discussing results in the order that they have been presented e.g. WT vs. TG, then ICV injection to ICR mice

• Line 245: “known to precede other processes like the…”

References

• Line 280 should be “References”

Minor Comments

• In my version, the figures look pixelated (e.g. text is hard to read in Figure 3). Please check the resolution of all figures.

6. PLOS authors have the option to publish the peer review history of their article (what does this mean?). If published, this will include your full peer review and any attached files.

Reviewer #1: Yes: Taher Darreh-Shori

Reviewer #2: Yes: Rajib Kumar Dutta

Reviewer #3: No

---

## [Author Response · Author response to Decision Letter 0]

5 Dec 2019

We have a separate file upload for the response to reviewers.

---

## [Editor Report · Decision Letter 1]

26 Dec 2019

Alterations of aqueous humor Aβ levels in Aβ-infused and transgenic mouse models of Alzheimer disease

PONE-D-19-24249R1

Dear Dr. Kim,

We are pleased to inform you that your manuscript has been judged scientifically suitable for publication and will be formally accepted for publication once it complies with all outstanding technical requirements.

With kind regards,

Madepalli K. Lakshmana, Ph.D

Academic Editor

PLOS ONE
---

## [Editor Report · Acceptance letter]

31 Dec 2019

PONE-D-19-24249R1 

Alterations of aqueous humor Aβ levels in Aβ-infused and transgenic mouse models of Alzheimer disease 

Dear Dr. Kim:

I am pleased to inform you that your manuscript has been deemed suitable for publication in PLOS ONE. Congratulations! Your manuscript is now with our production department. 

With kind regards,

on behalf of

Dr. Madepalli K. Lakshmana 

Academic Editor

PLOS ONE